# Nematode Pheromones: Structures and Functions

**DOI:** 10.3390/molecules28052409

**Published:** 2023-03-06

**Authors:** Biyuan Yang, Jie Wang, Xi Zheng, Xin Wang

**Affiliations:** State Key Laboratory for Conservation and Utilization of Bio-Resources in Yunnan, Yunnan University, Kunming 650091, China

**Keywords:** nematode pheromones, ascarosides, structures, functions

## Abstract

Pheromones are chemical signals secreted by one individual that can affect the behaviors of other individuals within the same species. Ascaroside is an evolutionarily conserved family of nematode pheromones that play an integral role in the development, lifespan, propagation, and stress response of nematodes. Their general structure comprises the dideoxysugar ascarylose and fatty-acid-like side chains. Ascarosides can vary structurally and functionally according to the lengths of their side chains and how they are derivatized with different moieties. In this review, we mainly describe the chemical structures of ascarosides and their different effects on the development, mating, and aggregation of nematodes, as well as how they are synthesized and regulated. In addition, we discuss their influences on other species in various aspects. This review provides a reference for the functions and structures of ascarosides and enables their better application.

## 1. Introduction

Nematodes are the most abundant animals on Earth, and they can be found virtually everywhere on land and in water [1]. Nematodes have a variety of lifestyles; some nematodes live freely, such as *Caenorhabditis elegans*, whereas some nematodes show parasitic lifestyles, such as root-knot nematodes (RKNs) and cyst nematodes (CNs). The impacts of nematodes extend to various domains of life. Plant parasitic nematodes cause a destructive loss in crop productivity, and, on the contrary, entomopathogenic nematodes can be used to kill insect pests in agriculture [2,3]. There are numerous chemicals that are involved in all aspects of nematode communication and life, mainly pheromones.

Pheromones are chemicals or mixtures of chemicals that can function as communication agents. The use of pheromones is very widespread in nature, for example, in protozoan ciliates, pheromones have functions in self/nonself recognition, vegetative reproduction, and mating interactions [4,5]. The fall armyworm *Spodoptera frugiperda* can accurately monitor its field population dynamics using sex pheromones [6]. In the locusts *Locusta migratoria*, only the existence of gregarious male adults can stimulate the synchronization of the sexual maturity of female adults. Among a large number of volatiles released by gregarious male adults, the aggregation pheromone 4-vinyl anisole is considered to play a key role in inducing the synchronization of female sexual maturity [7]. Using this function allows for better control of insect population densities, which also helps to protect crop yields. Pheromones also play a very important role in vertebrates. For example, adult sea lampreys *Petromyzon marinus* release large quantities of bile acid pheromones that attract mature females [8]. The pheromones (*Z*)-7-dodecenyl acetate and frontalin have been found in Asian elephants and have specific effects on elephant sexual behavior [9].

Moreover, pheromones also influence nematodes in various ways [10,11]. Unlike other animal species, ascarosides are the main type of pheromone produced by nematodes. These ascarosides comprise fatty-acid-derived side chains attached to the 3,6-dideoxysugar L-ascarylose, and their C-terminus or four loci can be modified [12]. Interestingly, these glycolipids are found almost exclusively in nematodes, including free-living nematodes and nematode species that parasitize insects, vertebrates, and plants [13]. In addition, a few other small molecules function as pheromones in nematodes, but there are relatively few reports on these molecules. Nematode pheromones are capable of regulating many aspects, including their development, mating, aggregation, and many others.

To date, several different nomenclatures have been used for ascarosides. First, one nomenclature system combines the functions of the compounds. For instance, the dauer-inducing ascr#1, ascr#2, and ascr#3 have been referred to as “daumone-1”, “daumone-2”, and “daumone-3”. Second, there is a nomenclature based on chemical structure. For example, according to the length of the carbon side chains on ascarylose sugar, the seven-carbon ascr#1 was named “C7” and the three-carbon ascr#5 was named “C3” [14]. However, as an increasing number of ascaroside structures have been identified, a third nomenclature (small molecule identifiers, SMIDs) has been adopted to name nematode metabolites. The overall structural class of a compound is denoted by four lowercase, non-italic letters in SMIDs, and a pound sign and a number are included, e.g., ascr#1 and icas#9. All assigned SMIDs can be found in the SMID database (www.smid-db.org) [15]. This third nomenclature no longer refers to the function or molecular structure of ascarosides, but to the order of their discovery, for example, ascr#1, 2, 3 … n, and then distinct abbreviations are used for ascarylose-containing molecules that contain additional moieties, such as when the sugar is decorated with an indol (icas#), octopamine succinate (osas#), hydroxybenzoyl (hbas#), or methyl-butenoyl (mbas#) [16].

The pheromones secreted by nematodes play an integral role in their communication and social behaviors. The research on nematode pheromones not only facilitates the use of pheromones for biological control but also serves as a useful reference for understanding the structures and functions of pheromones in the future. The current research has focused on model organisms such as *C. elegans*, and comparatively little research has been conducted on other species of nematodes. In this paper, we review the structures and functions of different nematode pheromones.

## 2. Synthesis and Regulation of Ascarosides

Many primary metabolic pathways participate in the synthesis of ascarosides, including the tricarboxylic acid cycle, amino acid catabolism, the peroxisomal β-oxidation of long-chain fatty acids, etc. There are four genes (*dhs-28*, *acox-1*, *daf-22,* and *maoc-1*) and complex signaling pathways (steroid hormones, serotonin, cGMP, TGF-β, insulin/IGF signaling, etc.) that are involved in ascaroside synthesis in *C. elegans* [11,14,17,18]. It has been shown that acyl coenzyme A oxidase ACOX-1, enoyl coenzyme A hydratase MAOC-1, β-hydroxyacyl-CoA dehydrogenase DHS-28, and β-ketoacyl -CoA thiolase DAF-22 primarily act in each step of the β-oxidation cycle [19]. The ACOX-1 encoded by the *acox-1* gene is the main enzyme for the synthesis of ascarosides in *C. elegans* [20]. It has fatty acid oxidation activity and interacts with the peroxide PEX-5 in peroxidase bodies [21]. The MAOC-1 is necessary for the biosynthesis of ascarosides’ fatty-acid-derived side chains via peroxisomal β-oxidation, and *maoc-1* is the gene related to the regulation of this process [22]. Meanwhile, thiolase DAF-22, a down-regulating factor for the beta-oxidation of the *C. elegans* peroxidase body, represents a single gene in *C. elegans* and two genes (*Ppa-daf-22.1* and *Ppa-daf-22.2*) with different domains in the free-living nematode *Pristionchus pacificus* [23]. Under conditions of adequate nutrition, the biosynthesis of ascarosides is carried out only by Ppa-daf-22.1. In contrast, Ppa-daf-22.2 is induced in the absence of food, leading to the production of specific ascarosides [23]. *Dhs-28* encodes a homolog of human D-bifunctional protein that functions upstream of SCPx and is also necessary for pheromone production [24]. Finally, the related reactions regulated by these four genes greatly affect the synthetic process of ascaroside pheromones in *C. elegans*.

Ascaroside pheromones can perform their biological functions by modulating signaling pathways participating in neuronal transmission. The main pathways for metabolizing ascarosides are as follows: the GPCR-Gqα [25], DAF-7/TGF-β [26], MAPK [27], DAF-2/Insulin pathways, etc. [28]. These pathways also participate in the adjustment of many neurons, mainly including the neurons AWA, ASH, ASI, and ADL and male-specific CEM neurons [11,14,17,18,29,30]. These neurons mainly regulate physiological activities together with related receptor genes. For example, *C. elegans* hermaphroditism, which acts mainly through self-fertilization, increases the mating rate in males after pathogen exposure, and this increase requires *str-44* in AWA neurons [31]. During the induction of ascr#10, *C. elegans* triggers the ADL sensory neuron process for signal transduction; this process triggers the main *mod-1* receptor to respond [32]. When ascarosides act on adults, they attenuate the expression of the insulin peptide INS-6 in ASI chemosensory neurons, resulting in a decrease in neuroendocrine insulin signals, which in turn prolongs reproductive duration [33]. The *crh-1* gene of *C. elegans* autonomously functions in ascr#5-sensing ASI neurons and inhibiting the formation of L2d [26]. In *C. elegans*, the tyra-2 receptor (a neurotransmitter-sensitive G-protein-coupled receptor) in ASH cells of nociceptive neurons is involved in the induction of osas#9 avoidance expression [34]. Ascarosides reversibly inhibit the expression of the *str-3* chemoreceptor gene in ASI neurons. At the same time, the suppression of *str-3* requires the involvement of the pheromone receptors SRBC-64/66 and SRG-36/37 [35]. However, more work is needed to clarify the specific mechanisms of these processes in nematodes.

## 3. Pheromones Secreted by Nematodes

Ascarosides (ASCRs) represent the majority of the pheromones secreted by nematodes. The molecular formula for an ascaroside, C_33_H_68_O_4_, was first proposed by Schulz and Becker in 1933. In 1957, Fairbairn et al. determined the structural formula for ascarosides. Through more in-depth studies on nematode pheromones, it was found that ascaroside derivatives, such as indole ascarosides (ICASs) and the ω-1 oxidation isomers of ASCRs, named OSCRs, can also act as pheromone components. Different phenotypes of nematode species are produced by different ascarosides or combinations of ascarosides; even slight changes in the chemical structure tend to produce drastically different patterns of activity. As a rule, the patterns of the biosynthesis of ascarosides are linked to the phylogeny, lifestyle, and ecological niche of the organism [14,36,37]. In addition, different concentrations of the same ascarosides can have different effects on nematodes. Other chemicals such as vanillic acid function as pheromones in some nematodes, but there have been comparatively few studies and discoveries in this area [38].

### 3.1. Development-Related Pheromones Secreted by Nematodes

The ability of nematodes to be so widely distributed in nature is closely related to their special developmental patterns. *C. elegans* lives freely in soil; it has a small, transparent body and serves as a model nematode species [39]. When the environment is suitable, an individual *C. elegans* starts to develop from a fertilized egg and progresses to adulthood through four stages of development. However, it stops feeding and developing if it encounters extreme conditions, such as a lack of food, elevated temperatures, or an increase in population density, and then the larva may enter a highly stress-resistant state called dauer diapause. This stage can last for several months. The nematodes eventually resume development and molt into the reproductive cycle under suitable conditions [40,41,42,43,44]. Much research shows that chemical pheromones can control dauer entry and exit [45,46].

The first dauer-inducing pheromone (daumone) was identified by means of the ethyl acetate extraction of a *C. elegans* liquid medium. The molecular structure of daumone was thereby determined to be (2)-(6*R*)-(3,5-dihydroxy-6-methyltetrahydropyran-2-yloxy) heptanoic acid, abbreviated to ascr#1 (also called daumone-1/ascaroside C7/asc-C7) (Table 1) [47]. Subsequently, ascr#2 (also called daumone-2/ascaroside C6) (Table 1) and ascr#3 (also called daumone-3/asc-∆C9/ascaroside C9) (Table 1) were also isolated and identified. Ascr#2 and ascr#3 induce dauer formation about 100 times more potently than ascr#1 does [48]. The pheromones that induce dauer formation may be single ascarosides or mixtures of different ascarosides, and these pheromones often act synergistically when mixed together. The dauer pheromone of *C. elegans* is mainly composed of ascr#2, ascr#3, and several other components, while in *Caenorhabditis briggsae*, the main component of the dauer pheromone is ascr#2 [49]. A derivative of ascr#2 has a β-glucosyl substituent linked to C2 of the ascarylose in ascr#2 and is named ascr#4 (also called daumone-4) (Table 1). The activity of ascr#4 is low [50]. Ascr#5 (also called daumone-5/ascaroside C3/asc-ωC3) (Table 1) is a potent dauer pheromone. The main function of ascr#5 is the activation of the axon regeneration pathway via SRG-36/SRG-37 GPCRs and EGL-30, indicating ascaroside signaling promotes axon regeneration by activating the GPCR-Gqα pathway [25]. In addition, the *crh-1* gene plays an autonomous role in ascr#5, sensing ASI neurons in order to inhibit the dauer formation of *C. elegans* L2d [26]. Ascr#5 also produces synergistic effects with ascr#2 and ascr#3 [51]. In ASI neurons, ascaroside pheromones (compounds composed of ascr#2, ascr#3, and ascr#5) reversibly inhibit the expression of the *str-3* chemical receptor gene, and when ascarosides are removed, its expression resumes. This process mainly occurs through the GPCR receptors SRBC-64/66 and SRG-36/37, which are required for *str-3* repression [35]. Ascr#8 (Table 1) uniquely possesses a p-aminobenzoate group in its terminus; this group is a folate precursor that is derived from bacteria and is not synthesized by *C. elegans* [52].

Furthermore, Butcher et al. [53] used activity-guided fractionation and NMR to discover a structurally novel indole-3-carboxylic acid-modified ascaroside in *C. elegans* named icas#9 (also called indolecarboxyl ascaroside C5/ascaroside C5/IC-asc-C5) (Table 1). It can induce dauer development at low (nanomolar) concentrations yet is inhibited at higher concentrations. Nacq#1 (Table 1) also acts antagonistically with respect to dauer-inducing ascarosides. The N-acylated glutamine derivative nacq#1 is mainly found in the excretions of males and contains an uncommon triply unsaturated ten-carbon fatty acid. Nacq#1 signals that enough resources are available to finish the dauer stage and resume reproductive growth. Although it reduces lifespan, nacq#1 can antagonize diapause and accelerate development, hastening sexual maturation [54]. It also has a trans-isomer, nacq#2 (Table 1) [54], but its function is unknown.

*P. pacificus* is a model species that has been extensively studied in biology [55]. This nematode can enter the dauer stage or other stages if food is enough for growth [56,57]. Typically, the mouth of an adult that preys on other nematodes is more complex than that of a bacterivorous nematode. Pheromones can regulate the mouth dimorphism of *P. pacificus* [58]. Neelanjan et al. [59] analyzed fractions of the *P. pacificus* exo-metabolome and found that it has rich signaling molecules controlling adult phenotypic plasticity, including ascarosides ascr#1, 9, and 12. Pasc#9 (Table 1) was the most abundant derivative after pasc#1 and pasc#12. Pasc#9 comprises an N-succinyl 1-phenylethanolamide connected to ascarylose with a 4-hydroxypentanoic acid chain. Dasc#1 (Table 1) consists of two ascr#1 units; one ascr#1 unit is connected to carbon 4 of the other ascr#1 unit. A 3-ureido isobutyrate moiety is also present on carbon 4 of ubas#1 (Table 1), and ubas#1 also contains ascr#9 with the (ω)-oxygenated ascaroside oscr#9 connected at position 2 [59]. Recent studies revealed that the formation of ubas#1 and related metabolites specifically requires the putative carboxylesterase *Ppa-uar-1* [60]. Additionally, dimeric ascarosides and ureido isobutyrate-substituted metabolites were first reported in *P. pacificus.* L-paratose forms the basis of part#9 (Table 1). Part#9 only differs from ascr#9 in terms of the stereochemistry of one hydroxyl group. Part#9 is also one of the components of npar#1. Npar#1 (Table 1) contains a derivative of the nucleoside adenosine. Although part#9, npar#1, ubas#1, and pasc#9 can induce dauer formation, npar#1 has a more intense effect than the others. Pasc#9, ascr#1, dasc#1, and npar#1 can induce eurystomatous mouth formation, a predatory morphology in the final larval and juvenile stages, in which *P. pacificus*-specific dasc#1 plays an important role [61,62].

*Heterorhabditis bacteriophora* is parasitic toward insects and has a developmental process similar to that of *C. elegans*. In the soil, the infective juveniles (IJs) survive as the only state of entomopathogenic nematodes. After IJs infect the host insects, they recover and lay eggs in their adults, which develop through four larval stages (J1–J4) to form the next generation [63,64,65]. In this process, *H. bacteriophora* secretes the ascaroside C11 ethanolamine (asc C11 EA) (Table 1), which prevents IJs from recovering to the J4 stage. Asc C11 EA comprises an ascarylose sugar, an ethanolamine fragment, and a carbon side chain containing ω-1 alcohol; the fatty-acid-derived portion of the side chain is 11 carbons long. Asc C11 EA and the dauer pheromone of *C. elegans* show structural similarity [66].

Figure 1 illustrates the schematic structure of the pheromones secreted by nematodes during their development. The figure presents a schematic diagram showing ascarylose sugars, variable-length fatty acids, and other moieties modifying them, which form different species of development-related ascaroside species.

### 3.2. Sex Pheromones Secreted by Nematodes

Mate selection is universal in sexually reproducing organisms, and pheromones provide individuals with advantageous mating information that helps them to select high-quality mates. In the twentieth century, the first sex pheromone was named bombykol, which is released from female silk moths (*Bombyx mori*) [67]. Sex pheromones have since been researched in more depth; they are defined as chemical substances produced by individuals that cause innate and rigid sexual behavior [68]. These pheromones have both sex- and species-specific effects. Nematode mating behavior is also regulated by pheromones [69]. Generally, the nematode mating response can be induced when the pheromone concentration is much lower than the concentration required for dauer formation.

*C. elegans* mainly reproduces as a hermaphrodite. However, most *Caenorhabditis* worm species achieve this by means of cross-fertilization. These hermaphrodites are essentially females with the ability to self-fertilize, and they can also mate with males, but their numbers are typically relatively low. Hermaphrodites do not appear to be attracted to male *C. elegans*, but males are attracted to them [70]. The ascarosides ascr#2, 3, 4, and 8 (Table 1) not only play roles in regulating nematode development but also function as sex pheromones that are known to attract males [71,72]. They show synergistic effects, whereby a mixture of ascr#2, 3, and 4 is an effective male attractant at low concentrations. Ascr#3 attracts *C. elegans* males but repels hermaphrodites and can increase the lifespan of *C. elegans*. Ascr#8 is a strong male-specific attractant and shows synergy with ascr#2 and ascr#3. A mixture composed of ascr#3 and ascr#8 strongly attracts males at ultra-low concentrations, but at higher concentrations, it is repulsive to hermaphrodites [36,50,71]. The other two ascarosides with sex pheromone functions, ascr#6.1 (Table 1) and ascr#6.2 (Table 1), were identified by Paul as diastereomeric side-chain-hydroxylated ascarosides [71]. Ascr#10 (also called asc-C9) (Table 1) makes up the majority of the sex-specific milieu of ascarosides produced by male *C. elegans*. Ascr#3 has an α, β-unsaturated fatty acid moiety, whereas ascr#10 has the corresponding dihydro-derivative; such minor structural modifications deeply influence their signaling properties. The male pheromone ascr#10 strongly attracts hermaphrodite nematodes and shortens their lifespan [73,74,75]. It also can increase germline proliferation and physiological cell death [76] and change the reproductive physiology of hermaphroditism, such as by improving sperm orientation and increasing the number of reproductive precursor cells in adults [77,78,79]. Furthermore, Dong et al. conducted a comparative analysis of indole ascaroside signaling for 14 *Caenorhabditis* species. Icas#2 and icas#6.2 (Table 1) were isolated from hermaphrodites of *C. briggsae* and were found to synergistically attract conspecific males [80].

*Panagrellus redivivus* has an ecological niche similar to that of *C. elegans*; it has a free-living lifestyle but belongs to a different clade. In contrast with *C. elegans*, the virgin females of *P. redivivus* attract and are attracted by the males, but they do not attract the same sex [81]. The ascaroside biosynthesis in *P. redivivus* is highly sex-specific. The females of *P. redivivus* can excrete ascr#1, ascr#10, and bhas#10 (Table 1) [74]. The males of *P. redivivus* can excrete dhas#18 (Table 1) [74]. Ascr#1 can strongly attract males, but high concentrations of ascr#1 repel the females of *P. redivivus*. At high concentrations, bhas#10 and ascr#10 attract males rather than females. Dhas#18, which is a known dihydroxy derivative of ascr#18 secreted by males as well as an ascaroside with extensive functionality as a characteristic of its lipid-derived side chain, can strongly attract the females of *P. redivivus*. Bhas#18 (Table 1) is a precursor for dhas#18 synthesis, but its exact function is unclear [74].

*Rhabditis* sp. SB347 is a unique free-living dioecious species that is often used in the laboratory [82]. The females of SB347 produce ascr#1 and ascr#9 (Table 1), which function as sex pheromones. At femtomolar levels, ascr#1 and ascr#9 are strongly attracted to males, but not to hermaphrodites and female nematodes [83].

In addition to the ascaroside pheromones, a different type of pheromone is secreted by *Heterodera glycines*, a plant nematode that is parasitic toward soybeans. The females secrete vanillic acid (Table 1), which also functions as a sex pheromone [38,84]. However, there are very few reports on non-ascarosides acting as pheromones in nematodes.

The female beet cyst nematode *Heterodera schachtii* can excrete a sex pheromone. The pheromone consists of at least two components, and the pheromone component is soluble in aqueous solutions with diethyl ether. These components may show superposition rather than synergy. However, their exact structure is unclear [85].

*Bursaphelenchus xylophilus* is a pine wood nematode (PWN). It has been shown that both sexes of *B. xylophilus* produce sex pheromones: unmated females attract conspecific males, and males attract both mated and unmated females through volatile chemical compounds. Additionally, *Bursaphelenchus okinawaensis*, which is associated with insect vectors and host plants, produces a pheromone that attracts males [86,87]. However, the exact composition of the compound is unknown.

The dimorphism of the adults is an important feature of the life history of *Globodera rostochiensis*. Hermaphrodites attract males for mating by producing pheromones. Four fractions of the homospecific sex pheromone produced by virgin females, which were isolated using chromatography technology, were tested for their ability to attract male *G. rostochiensis*; only two of the fractions showed sex pheromone activity. Several weakly basic polar compounds constitute the sex pheromone of *G. rostochiensis*. The exact structure of its components is unclear [88].

The chemical structure diagram for sex pheromones secreted by nematodes is shown in Figure 2. This diagram shows ascaroside building blocks associated with the mating of different species of nematodes, including ascarylose sugars, variable-length fatty acids, and other modification groups.

### 3.3. Aggregation of Pheromones Secreted by Nematodes

*C. elegans* uses specifically modified forms of the ascarosides that contain indole units as highly effective aggregation pheromones. The indole ascarosides (ICASs) incorporate an L-tryptophan-derived indole-3-carboxylic acid group, which is linked to the four-position of the ascarylose moiety. An indole carboxy unit forms one indole derivative, and it is connected to an ascarylose bearing a nine-carbon unsaturated side chain identical to that found in the known ascr#3; this indole carboxy ascaroside is called “icas#3” (Table 1). The icas#3 occurs primarily by means of an expression protein in *C. elegans*, CEST-3, adding an IC group to the corresponding unmodified ascr#3 [15,89]. Icas#3 and icas#9 are relatively good attractants [15]. The 4-hydroxybenzoyl derivative of ascr#3 is called hbas#3 (Table 1). Hbas#3 was the first ascaroside with a 4-hydroxybenzoyl structure to be discovered. Hbas#3 strongly attracts *C. elegans* at low concentrations (10 fM), more effectively so than icas#3 and icas#9 [19]. Ascr#5 in combination with ascr#2 or ascr#3 may influence the aggregation of *C. elegans* adults; however, more in-depth research is needed on this topic [90].

See Figure 3 for a schematic overview of nematode pheromones related to their aggregation. It illustrates ascarylose sugars, fatty acids with variable lengths, and other modifications that form aggregation-related ascaroside species.

### 3.4. Pheromones with Other Functions Secreted by Nematodes

The L1 larvae of *C. elegans* can specifically produce certain octopamine ascarosides, in which the ascarylose four-position is linked to a side chain derived from the succinylation of the neurotransmitter octopamine. The octopamine ascarosides osas#2 (Table 1), osas#9 (Table 1), and osas#10 (Table 1) play roles in dispersal [72]. Osas#9 is a pheromone that acts as a dispersal signal, especially in the case of a lack of food. Avoidance reactions to osas#9 require the G-protein-coupled receptor TYRA-2 [34]. With a continuous decrease in food, ascr#10 and osas#10 are converted to ascr#9, osas#9, and icas#9 [72].

Ascr#3 was found to regulate metabolism and avoidance behavior, it was defined as a population density pheromone. When food is scarce, ascr#3 causes hermaphrodites to have an avoidance effect [91,92,93]. When the ADF of a single sensory neuron is removed, both sexes are weakly rejected by the ascaroside ascr#3. Although ADF has functions in both sexes, ascr#3 is only detected in males, which is the result of the main sex regulator *tra-1* [94]. A derivative of ascr#3 called mbas#3 (Table 1) was the first ascaroside discovered to have an (E)-2-methyl-2-butenoyl structure [19], and it acts as a dispersal signal in *C. elegans*, as well as having an antagonistic effect on the attractant characteristics of indole ascarides such as icas#3 and icas#9 [95].

*C. elegans* exist in two states (roaming and dwelling) when searching for bacterial food [96]. At physiological levels, some ascarosides regulate foraging by inhibiting roaming behavior. Ascr#2, 3, 5, and 8 and icas#9 have some effects on the foraging behavior of *C. elegans*. Different strains of *C. elegans* have different sensitivities to icas#9 due to differences in the expression of the *srx-43* gene, which encodes the icas#9 receptor [97]. *C. elegans* can also develop certain memory behaviors in response to pheromones such as ascr#3, ascr#5, and icas#9 [98,99,100].

Research has shown that abundant ascr#18 (Table 1) could be isolated from *Meloidogyne incognita*, *Meloidogyne javanica*, and *Meloidogyne hapla*, as well as from cyst (*H. glycines*) and lesion (*Pratylenchus brachyurus*) nematodes [101]. This compound is also present in *C. elegans* [19] and entomopathogenic nematodes [36,66]. Ascr#18 can be sensed by a wide range of plant species, which in turn mount defense responses against nematodes.

Entomopathogenic nematode IJs can sense ascaroside mixtures including ascr#2, 3, and 8 and icas#9 from *C. elegans*, which causes the dispersal of nematodes. The pheromone mixture of ascr#9 and ascr#11 (Table 1) from consumed insect host corpses contributes to IJ dispersal [102,103]. Ascr#9 and ascr#11 are structural analogs, and they can be interchanged in the mixture of dispersal pheromones. Ascr#9 was detected in some species of *Steinernema* spp. (*S. feltiae*, *S. carpocapsae*, *S. riobrave*, and *S. diaprepesi*) and *Heterorhabditis* spp. (*H. zealandica*, *H. floridensis,* and *H. bacteriophora)*, which indicated that ascr#9 may be widely present in dispersal mixtures from entomopathogenic nematodes. However, it was found that ascr#11 is present in some species of *Steinernema*, but not in *Heterorhabditis*, indicating that ascr#11 may be specific to *Steinernema* [103,104]. Ascr#12 (Table 1) can induce the IJs recovery of *H. bacteriophora* H06 [105], while ascr#11 can enhance the IJ yields of *Steinernema carpocapsae* All and *H. bacteriophora* H06 in the liquid medium [106].

The pinewood nematode, *B. xylophilus*, the causal agent for pine wilt disease and a global quarantine pest, usually displaces *Bursaphelenchus mucronatus*, a native sympatric sibling species. Similar to what occurs in *C. elegans*, in *B. xylophilus*, the pheromones comprise the hydrophilic ascarosides family, which are derivatives of 3,6-dideoxy-L-saccharose linked to fatty-acid-derived side chains; they regulate the transmission of *B. xylophilus* and its vector beetle, and they regulate the lifecycle of *B. xylophilus* [107]. Ascr#9 is the major component of the ascarosides of the two nematodes; it not only increases the number of invasive strains but also reduces the number of native strains [108]. Moreover, ascr#9 plays a leading role in pheromone-regulated reproductive plasticity. At the molecular level, two genes, *Bxydaf-38* and *Bxysrd-10*, participate in the perception of ascr#9 [108]. When mixed with ascr#12, it acts synergistically and also increases body length in the females of *B. xylophilus*, though it reduces body length in *B. mucronatus* [109]. 

*C. elegans* has shown a tendency to be attracted to a series of odorous substances, and with the passage of time, this tendency changes from attraction to dispersal [98,99]. This varied pheromone-mediated behavior is called olfactory plasticity, which depends on the population density [110]. However, the pheromone component that plays a major role in this process has not been identified. In addition, the pheromones released by injured conspecific nematodes are repellent to nematodes, and they may contain alarm pheromones. These alarm pheromones may not belong to the ascaroside class of pheromones [111]. However, their exact structure has not been identified.

The following Figure 4 displays the chemical structure diagram of function pheromones secreted by nematodes. This diagram shows the ascaroside building blocks associated with the functions of different species of nematodes, including ascarylose sugars, variable-length fatty acids, and other modification groups.

**Table 1 molecules-28-02409-t001:** Pheromones are secreted by nematodes.

Name	Chemical Constitution	Function	Organism	Reference
Asc C11 EA	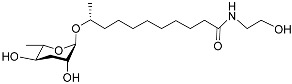	Development	*C. elegans*	[66]
Ascr#1	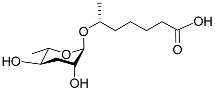	Development; mating	*C. elegans*, *P. pacificus*, *P. redivivus*, and *Rhabditis* sp. SB347	[47,48,59,61,62,74,83]
Ascr#2	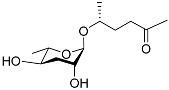	Development, mating, foraging, and dispersal	*C. elegans*; *C. briggsae*	[48,49,71,72,97]
Ascr#3	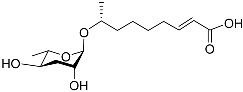	Development, mating, foraging, and dispersal	*C. elegans*	[36,48,49,50,71,72,90,91,92,93,95,97,98,99,100]
Ascr#4	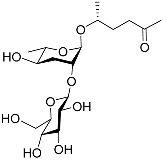	Development; mating	*C. elegans*	[50,71,72]
Ascr#5	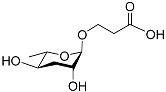	Development; foraging	*C. elegans*	[25,26,51,97]
Ascr#6.1	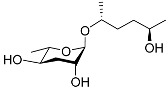	Mating	*C. elegans*	[71]
Ascr#6.2	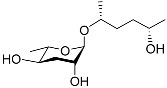	Mating	*C. elegans*	[71]
Ascr#8	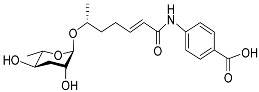	Development, mating, foraging, and dispersal	*C. elegans*	[36,50,52,71,72,97]
Ascr#9	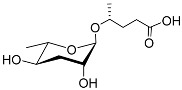	Mating; dispersal	*C. elegans*, *P. pacificus*, *Rhabditis* sp. SB347, *B. xylophilus*, *B. mucronatus* *H. bacteriophora*, *H.zealandica*, *H. floridensis*, *S. carpocapsae*, *S. riobrave, S. diaprepesi*, and *S. feltiae*	[36,59,83,102,103,104,108,109]
Ascr#10	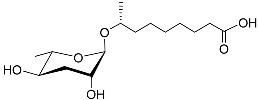	Mating	*C. elegans*; *P. redivivus*	[73,74,75,76,77,78,79,101]
Ascr#11	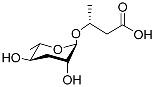	Dispersal	*C. elegans*, *S. carpocapsae*, *S. riobrave*, *S. diaprepesi*, and *S. feltiae*	[102,103,104,106]
Ascr#12	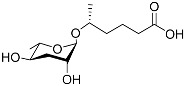	Development	*C. elegans*; *P. pacificus*	[36,59,105,109]
Ascr#18	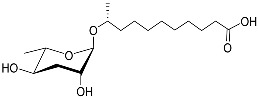	Avoidance	*M. incognita*, *M. javanica, M. hapla*, *H. glycines*, and *P. brachyurus*	[19,101]
Bhas#10	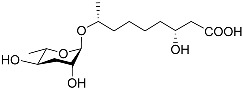	Mating	*C. elegans*; *P. redivivus*	[74]
Bhas#18	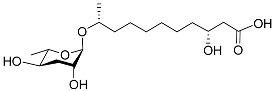	Unknown	*P. redivivus*	[74]
Dasc#1	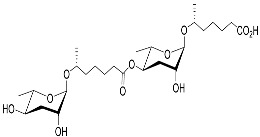	Development	*P. pacificus*	[59,61,62]
Dhas#18	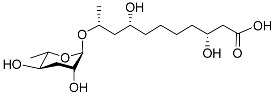	Mating	*P. redivivus*	[74]
Hbas#3	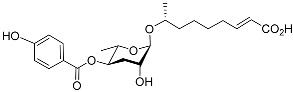	Aggregation	*C. elegans*	[19]
Icas#2	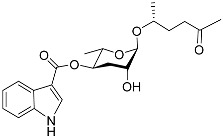	Mating	*C. briggsae*	[80]
Icas#3	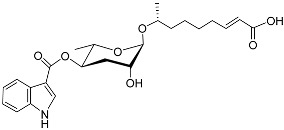	Aggregation	*C. elegans*	[15,89]
Icas#6.2	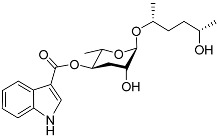	Mating	*C. briggsae*	[80]
Icas#9	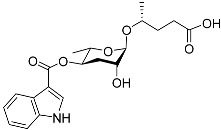	Development, aggregation, foraging, and dispersal	*C. elegans*	[15,19,53,72,97]
Mbas#3	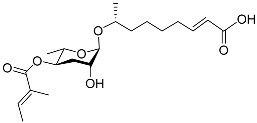	Dispersal	*C. elegans*	[19,95]
Nacq#1	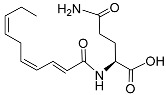	Development	*C. elegans*	[54]
Nacq#2	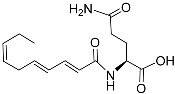	Unknown	*C. elegans*	[54]
Npar#1	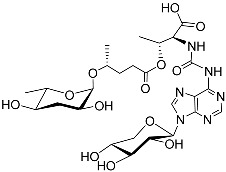	Development	*P. pacificus*	[59,61,62]
Osas#2	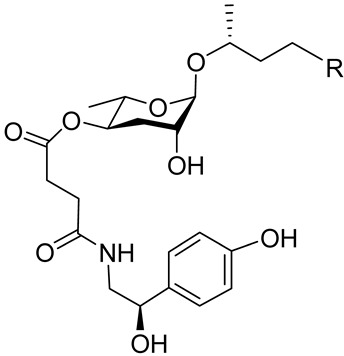	R=(C=O)CH_3_	Dispersal	*C. elegans*	[72]
Osas#10	R=(CH_2_)_4_COOH	Dispersal	*C. elegans*	[72]
Osas#9	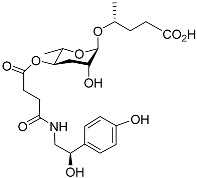	Dispersal	*C. elegans*	[34,72]
Part#9	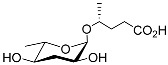	Development	*P. pacificus*	[59,61,62]
Pasc#9	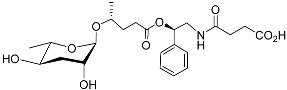	Development	*P. pacificus*	[59,61,62]
Ubas#1	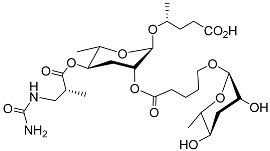	Development	*P. pacificus*	[59,61,62]
Vanillic acid	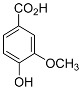	Mating	*H. glycines*	[38,84]

## 4. Nematode Pheromone Communication with Other Species

Nematode pheromones mainly function in an intraspecies manner, but further research has shown that they also function between different species [16,112], such as fungi, plants, and insects.

Manosalva et al. [101] found that nematode pheromones trigger defense responses in different organs of plants. Moreover, plants can metabolize nematode pheromones via peroxisomal β-oxidation and thus alter their chemical information, and they can produce a blend of ascarosides to control plant nematodes and reduce harm to themselves [13]. Interestingly, increased callose buildup was seen in *Arabidopsis* leaves after treatment with ascr#1 and ascr#18. AOS, PR1, PDF1.2, LOX2, and other defense-related genes also increased their expression as a result of ascr#18, which may have contributed to the improved plant defensive responses [113].

The typical pine wilt disease encompasses complex associations between PWN, symbiotic fungi, and vector beetles. In this system, nematode pheromones not only increase the number of mycelia, and improve the spread of fungi and nematode efficiency [114], but they also promote the pupation of beetles by inducing them to produce the molting hormone and upregulating the expression of genes related to the molting hormone [107]. Interestingly, PWN vector beetles can also produce ascarosides that promote the aggregation of their symbiotic plant-parasitic nematode species [107]. This indicates that nematode pheromones can regulate interspecific interactions.

Nematode-trapping fungi are predators that can consume nematodes and are widespread in soils of distinct ecological provenances [115]. Nematode-trapping fungi can detect and respond to nematode pheromones for the generation of trapping devices to catch and consume nematodes. For instance, the model *Arthrobotrys oligospora* can form traps of adhesive nets via the stimulation of ascaroside pheromones [27,116,117].

Recent studies have revealed that the main components of nematode pheromones, ascarosides, can be widely metabolized by animals, plants, and microorganisms [112], which may interact with certain nematodes by manipulating ascaroside signaling. The responses of other species to nematode pheromones may accelerate the rapid evolution of pheromones and may provide evidence for the synergistic evolution of species.

## 5. Conclusions

Pheromones play wide-ranging roles in nematodes, such as in their development, mating, aggregation, olfactory plasticity, and dispersal. These pheromones are closely related to a variety of factors such as lifestyle, sex, and developmental stage, and the nematodes living in various habitats can produce rich and diverse pheromones (Figure 5) [118,119]. *C. elegans* can secrete many kinds of ascarosides to improve development and induce dauer formation. Entomopathogenic nematodes can secrete different ascarosides to assist them in finding hosts and thriving. The pheromones secreted by plant parasitic nematodes are even closely related to interspecific competition. The latest research has shown that secreted pheromones can be sensed and even metabolized by organisms in the environment (such as animals, plants, and microorganisms).

Ascarosides are major components of nematode pheromones; they are highly conserved and species-specific, and the same ascarosides may play different roles among nematodes. In chemical terms, they comprise dideoxysugar ascaryloses linked to different fatty-acid side chains along with derivatives of amino acids, folate, and other primary metabolites. Structural and functional diversity exists due to differences in the lengths of the side chains and the derivatives. The effects of ascarosides on nematodes are not only highly dependent on their chemical structure but are also linked to their concentrations and the synergistic effects that take place between ascarosides [120,121].

Approximately 200 ascarosides have been discovered and identified from over 20 different nematodes [17,42,113]. The functions of most of them are unknown, whereas a few have been found to function as pheromones [19,122]. There are large interspecific differences in the structures and compositions of ascarosides [36]. However, these ascarosides and the *C. elegans* ascarosides share some structural similarities. For example, the ascarosides produced by *Ascaris suum* have long chain structures, similar to those of the ascarosides produced by *C. elegans* [71]. Recent research has shown that the nematode *C. briggsae* biosynthesizes ascarosides in a manner similar to *C. elegans* and also has a related developmental pathway that induces the stress-resistant dauer life stage. Thus, ascarosides may play similar roles in other nematode species compared with *C. elegans*. Studying the functions of ascarosides could provide a new method for controlling the parasitic nematodes, but they need to be further explored. Moreover, nematode pheromones have effects on other species, but the current study has revealed only the tip of the iceberg of the complex multidirectional communication network mediated by ascarosides. Therefore, it is important to further investigate the responses of other species to ascarosides, particularly regarding pathways and receptors, in order to explain this process.

The discovery of nematode pheromones provides new experimental channels for studying pheromone communication and its evolution, which will have significant value in the biological control of harmful parasitic nematodes and chemical ecology. A systematic description of the structures and functions of the pheromones of *C. elegans* and other nematodes will be helpful in improving our understanding of various biological processes. Although nematode pheromones are only a small class of compounds to be studied in depth, they will certainly have a major impact on the study of interspecific and intraspecific interactions in nematodes.

## Figures and Tables

**Figure 1 molecules-28-02409-f001:**
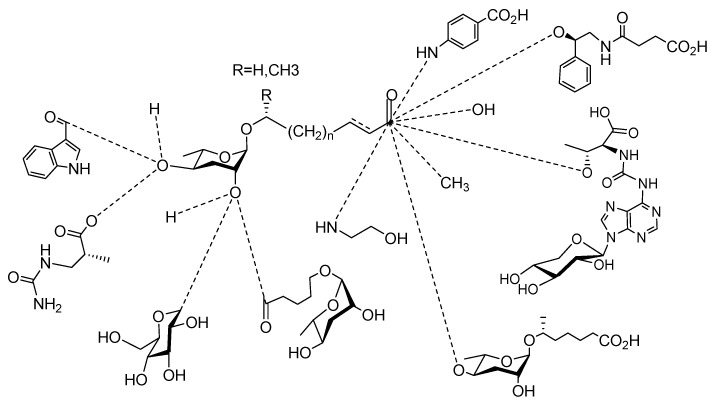
Overview of the chemical structure of development-related pheromones secreted by nematodes. Nacq#1 does not have an ascaroside structure, so it is not shown here, and its detailed structure is shown in Table 1.

**Figure 2 molecules-28-02409-f002:**
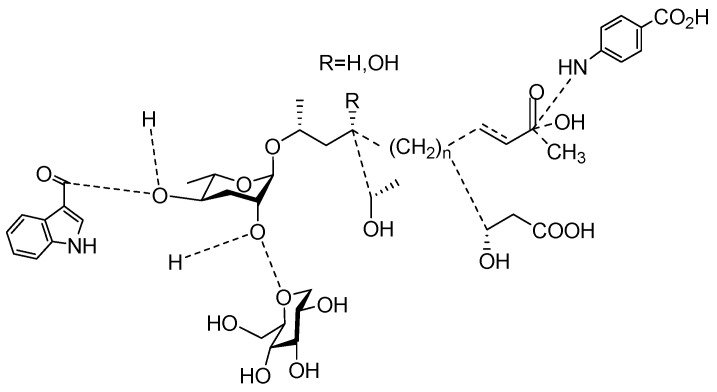
Overview of the chemical structures of sex pheromones secreted by nematodes. Vanillic acid does not have an ascaroside structure, so it is not shown here, and its detailed structure is presented in Table 1.

**Figure 3 molecules-28-02409-f003:**
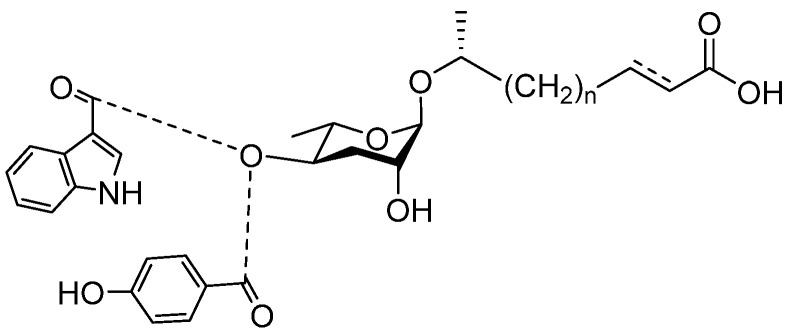
Overview of the chemical structure of the aggregation of pheromones secreted by nematodes.

**Figure 4 molecules-28-02409-f004:**
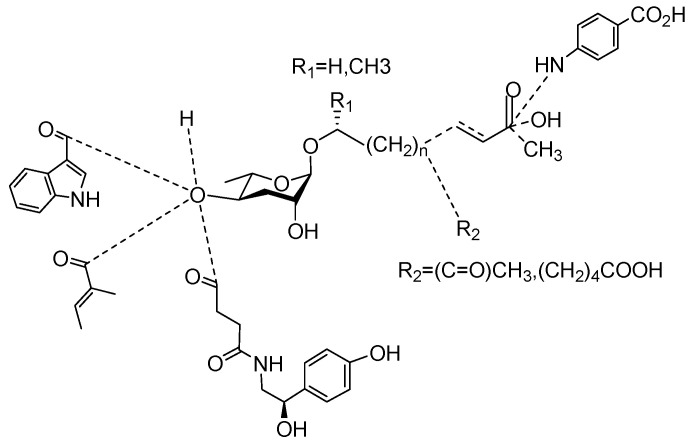
Overview of the chemical structures of pheromones with other functions secreted by nematodes.

**Figure 5 molecules-28-02409-f005:**
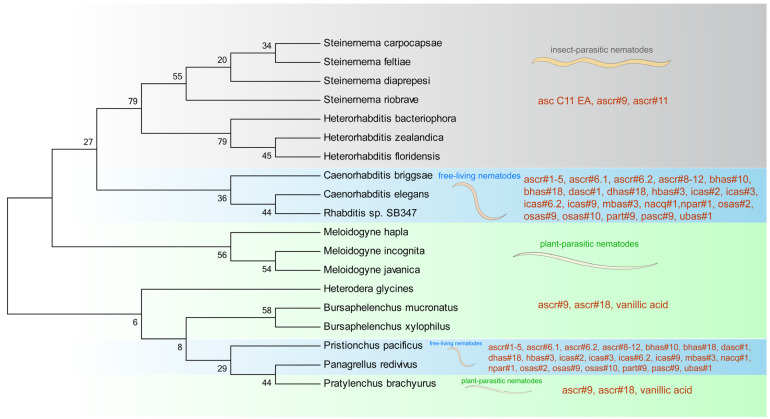
Pheromones are produced by a wide range of nematode species. The phylogenetic tree was drawn using Mega 4 software based on the comparison of 28S ribosomal RNA (28S) gene sequences obtained from GenBank. The nematodes can be classified into three categories including plant-parasitic nematodes (light green), insect-parasitic nematodes (light grey), and free-living nematodes (light blue). Diverse pheromones can be found widely among nematode groups (red).

## Data Availability

Not applicable.

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
