# Peer review of "Nematode Pheromones: Structures and Functions"

_molecules, 2023, doi:10.3390/molecules28052409_

Round 1
Reviewer 1 Report
The review by Yang et al. is directed to update information on molecular structures and activities of nematode pheromones (brought onto the scene by J.W. Golden & D.L. Riddle with their 1982 work “A pheromone influences larval development in C. elegans” in Science). This information is well worth being published on Molecules. However, NOT in the present form! As it is structured and presented, the review is practically unreadable and needs an in-depth reorganization. Also, it requires to be emended from (i) a lot of ‘naïve/poorly scientific’ sentences (starting from the first one in the Introduction, “Nematodes are a rich variety of species belonging to the phylum Nematoda”), (ii) wrong concepts/assertions (such as (Abstract) “Pheromones are chemicals secreted by organisms in vitro (?)…”, “This review … facilitates the discovery of novel nematode pheromones”, or (Introduction) “They [nematodes] are the most abundant Metazoa on Earth …”) , (iii) typing mistakes (such as “ascarisides” in Figure 1 legend), and (iv) wrong/misleading citations (such as “In the twentieth century, the German biochemist Adolf extracted the first sex pheromone, named bombykol, … [49]”; the right name is Adolf Friedrich Johann Butenandt (Nobel Prize in Chemistry in 1939), the article in question was published by Butenandt in 1959 in Z. Naturforsch., in collaboration with R. Beckmann, D. Stamm & E. Hecker, and in the reference list the number [49] wrongly indicates an article by “Gomez-Diaz, C.; Benton, R. The joy of sex pheromones. EMBO. Rep. 2013…”
In reorganizing their review, the Authors are strongly recommended, first of all, to consider that “Small molecule pheromones and hormones controlling nematode development” have exhaustively been reviewed not long ago (2017) by Butcher on Nat. Chem. Biol. Therefore, it would be appropriate to limit most attention on which molecular structures and activities have newly been described in the last 5-6 years in the field of nematode pheromones. By this restriction, not only the text would greatly improve conciseness and readability. The iconography would as well greatly gain comprehensibility and exhaustiveness. The four tables (awkwardly organized into three columns headed “Names”, Chemical constitution” and “References”) and the figure are overall extremely redundant, over-dimensioned (the figure extends on five pages!), largely overlapping in contents, and have no direct/clear integration with the text. The figure might be deleted and the four tables might well be concentrated into a single one organized with four columns reflecting the titles (“Development-related pheromones …”, “Sex pheromones …”, “Aggregation pheromones …”, “Pheromones with other functions …”) which are distinctive of each one. Second (but not less important), the Authors might/should (i) amplify (make more attractive for a wider readership) the Introduction section by adding some general information on what is known about structures and activities of pheromones from other eukaryotic organisms (in particular in ciliated protozoa, arthropods and vertebrates), (ii) provide throughout a wider and more systematic picture of the taxonomy, ecology and biology of the nematode species that are source of pheromones, and (iii) rename and re-modulate the last section (number 4) simply as ‘Concluding remarks’, or ‘Perspectives’ (considering that no result is therein actually discussed or interpreted, the name “Discussion” is largely out of the context).
Author Response
Dear reviewer,
On behalf of my co-authors, we appreciate reviewers very much for their positive and constructive comments and suggestions on our manuscript entitled “Nematode Pheromones: Structures and Functions” (Manuscript Number: 2216705). We revised the manuscript according to these comments and suggestions. In general, we have tried our best to revise our manuscript and provide the point-by-point responses. The following is a summary list of changes:
Point 1: A lot of ‘naïve/poorly scientific’ sentences (starting from the first one in the Introduction, “Nematodes are a rich variety of species belonging to the phylum Nematoda”).
Response 1: Thank you for pointing out this problem in the manuscript. This sentence is deleted in the revised manuscript. After reviewing the relevant literature we have deleted or modified other similar phrases in the text accordingly.
Point 2: Wrong concepts/assertions (such as (Abstract) “Pheromones are chemicals secreted by organisms in vitro (?)…”, “This review … facilitates the discovery of novel nematode pheromones”, or (Introduction) “They [nematodes] are the most abundant Metazoa on Earth …”) .
Response 2: Thanks for your suggestions. After revisiting relevant papers and materials, we reorganized the language to describe the relevant content, these sentences are modifieded in the revised manuscript.
Point 3: Typing mistakes (such as “ascarisides” in Figure 1 legend).
Response 3: Thanks for your suggestions. Figure 1 and related content has been removed. We have also added additional content and figure to the manuscript.
Point 4: Wrong/misleading citations (such as “In the twentieth century, the German biochemist Adolf extracted the first sex pheromone, named bombykol, … [49]”; the right name is Adolf Friedrich Johann Butenandt (Nobel Prize in Chemistry in 1939), the article in question was published by Butenandt in 1959 in Z. Naturforsch., in collaboration with R. Beckmann, D. Stamm & E. Hecker, and in the reference list the number [49] wrongly indicates an article by “Gomez-Diaz, C.; Benton, R. The joy of sex pheromones. EMBO. Rep. 2013…”.
Response 4: Thank you for pointing out this problem in the manuscript, we have modified relate name and reference.
Point 5: In reorganizing their review, the Authors are strongly recommended, first of all, to consider that “Small molecule pheromones and hormones controlling nematode development” have exhaustively been reviewed not long ago (2017) by Butcher on Nat. Chem. Biol. Therefore, it would be appropriate to limit most attention on which molecular structures and activities have newly been described in the last 5-6 years in the field of nematode pheromones. By this restriction, not only the text would greatly improve conciseness and readability. The iconography would as well greatly gain comprehensibility and exhaustiveness.
Response 5: Thanks for your suggestions! In reorganizing our review, we revisited the article “Small molecule pheromones and hormones controlling nematode development” and have added the relevant documents about the recent 5-6 years to the revised manuscript.
Point 6: The four tables (awkwardly organized into three columns headed “Names”, Chemical constitution” and “References”) and the figure are overall extremely redundant, over-dimensioned (the figure extends on five pages!), largely overlapping in contents, and have no direct/clear integration with the text. The figure might be deleted and the four tables might well be concentrated into a single one organized with four columns reflecting the titles (“Development-related pheromones …”, “Sex pheromones …”, “Aggregation pheromones …”, “Pheromones with other functions …”) which are distinctive of each one.
Response 6: Thank you so much for your comments! We have combined the four tables into one and enriched the table with content, such as added the function and the derived organism of each compound in the table. In view of the problems that appeared in the figure, we have removed the figure.
Point 7: Second (but not less important), the Authors might/should (i) amplify (make more attractive for a wider readership) the Introduction section by adding some general information on what is known about structures and activities of pheromones from other eukaryotic organisms (in particular in ciliated protozoa, arthropods and vertebrates).
Response 7: We have added about eukaryotic(such as ciliated protozoa, arthropods and vertebrates) pheromones content in the introduction section of the manuscript, including its chemical substances and functions.
Point 8: Provide throughout a wider and more systematic picture of the taxonomy, ecology and biology of the nematode species that are source of pheromones.
Response 8: We added to the manuscript a phylogenetic tree of pheromone-secreting nematodes, as well as their lifestyles and the corresponding pheromones they secrete. And added relevant content to the manuscript.
Point 9: Rename and re-modulate the last section (number 4) simply as ‘Concluding remarks’, or ‘Perspectives’ (considering that no result is therein actually discussed or interpreted, the name “Discussion” is largely out of the context).
Response 9: Thanks for your suggestions! “Discussion and Conclusions” is replaced by "Concluding remarks".
Once again, thank you very much for your comments and suggestions. And we hope that the revised manuscript can be accepted by molecules.
Thank you and best regards.
Sincerely,
Xin Wang
State Key Laboratory for Conservation and Utilization of Bio-Resources in Yunnan, Yunnan University, Kunming 650091, China
0086-15812045227
087165034838
xinwang2@ynu.edu.cn
Reviewer 2 Report
In this paper, Yang et al summarized the recent progresses on the effects of nematode ascarosides on the growth, mating and aggregation, as well as their impacts on other species. Generally, it’s an interesting topic which will attract wide audience among scientific communities. The reviewer will recommend it for the publication in Molecules after the following concerns being addressed.
1, To increase readability and clarity, the authors may want to include more information in the tables, such as the function for each compound, the derived organism of each compound. And the pheromones may be arranged in a more logic manner (the authors may want to place the same type of ascarosides together).
2, Given that the core structures of the nematode pheromones are the same, the reviewer suggested that a schematic summary of the structure-activity relationships of each function may be included in each section.
3, It’s suggested to keep consistent in drawing the structures of the pheromones in the tables throughout the manuscript, since some structures are blurry and some fragments are missing in several cases. It’s recommended to use Chemdraw with the ACS 1996 style to draw the structures.
4, It’s recommended to give a brief introduction of the nomenclature of the nematode pheromones, which can be referred to Frank C Schroeder’s work in https://www.smid-db.org/browse.
5, There also some English grammar error in the manuscript. A thorough proof reading and checking is needed before it can be resubmitted.
Author Response
Dear reviewer,
On behalf of my co-authors, we appreciate reviewers very much for their positive and constructive comments and suggestions on our manuscript entitled “Nematode Pheromones: Structures and Functions” (Manuscript Number: 2216705). We revised the manuscript according to these comments and suggestions. In general, we have tried our best to revise our manuscript and provide the point-by-point responses. The following is a summary list of changes:
Point 1: To increase readability and clarity, the authors may want to include more information in the tables, such as the function for each compound, the derived organism of each compound. And the pheromones may be arranged in a more logic manner (the authors may want to place the same type of ascarosides together).
Response 1: Thank you so much for your comments! We have added the function and the derived organism of each compound in the table. The pheromones in the table are arranged according to their initials, the same initials are sorted by their numerical size.
Point 2: Given that the core structures of the nematode pheromones are the same, the reviewer suggested that a schematic summary of the structure-activity relationships of each function may be included in each section.
Response 2: Thank you so much for your comments! We have add relate figure in each section. We combined the functions and structures of pheromones and drew a schematic overview of the structures of the relevant functional pheromones, at the same time, the specific structure can be found in Table 1.
Point 3: It’s suggested to keep consistent in drawing the structures of the pheromones in the tables throughout the manuscript, since some structures are blurry and some fragments are missing in several cases. It’s recommended to use Chemdraw with the ACS 1996 style to draw the structures.
Response 3: Thank you for pointing out this problem in the manuscript. We have used the drawing software "Chemdraw" to redraw the entire chemical structure and imported into our manuscript, to keep consistent in drawing the structures of the pheromones in the tables throughout the manuscript.
Point 4: It’s recommended to give a brief introduction of the nomenclature of the nematode pheromones, which can be referred to Frank C Schroeder’s work in https://www.smid-db.org/browse.
Response 4: Thanks for your suggestions. We have added nematode pheromone nomenclature to the introduction of the manuscript and have provided examples to help the reader understand it more easily.
Point 5: There also some English grammar error in the manuscript. A thorough proof reading and checking is needed before it can be resubmitted.
Response 5: Thanks for your suggestions. Prior to resubmission, we thoroughly re-proofed and rechecked the manuscript, any errors have been adjusted accordingly in the original text. We feel sorry for causing you unnecessary troubles in reviewing, we hope that the revised version may meet your exceptions.
Once again, thank you very much for your comments and suggestions. And we hope that the revised manuscript can be accepted by molecules.
Thank you and best regards.
Sincerely,
Xin Wang
State Key Laboratory for Conservation and Utilization of Bio-Resources in Yunnan, Yunnan University, Kunming 650091, China
0086-15812045227
087165034838
xinwang2@ynu.edu.cn
Round 2
Reviewer 1 Report
I take note that the Authors have substantially improved their manuscript, following in a large measure suggestions and criticisms provided in the reviewers’ comments. However, they have clearly disregarded the strong recommendation directed to revise the text with the help of a native English biologist. As it stands, the text (in a larger measure the parts that have been added and are highlighted in blue) still contains a wealth of awkward, grammatically incorrect and poorly constructed sentences that need to be removed. Also, it needs to be cleaned from a lot of ‘naïve’ expressions and wrong wording. The following is an exemplificative, scattered list of points that justify my criticism.
“Pheromones are chemicals that are secreted by the body”. “Nematodes, as one of the most abundant animals on Earth…”. “By contrast, another parasitic nematode, entomopathogenic nematodes can kill insects…”. “As an arthropod, Spodoptera exigua can …”. “In Locusta migratoria, which is also as an arthropod, …”. ]“Research on nematode pheromones not only facilitates the use of pheromones for biological control, but also serves as a useful reference for understanding the roles of pheromones.”. “… with a view to informing and inspiring subsequent researchers.” “In the beetle, miR-31-5p promotes production…” [The beetles/Coleoptera are thousand species!] “Ascr#12 is a pheromone, which can improve the recovery of H. bacteriophora H06 to promote development [110]. Surprisingly, it has been demonstrated that IJ yields were significantly improved in the liquid medium for Steinernema carpocapsae All and H. 420 bacteriophora H06 with proper concentrations (ascr#11).”
With regard to iconography, I remark that it is not possible to justify: (i) four repetitions of the same sentence to recall the four figures in the text, and (ii) the organization and the legend of Fig. 5.
“see Figure 1. The figure presents a schematic diagram showing ascarylose sugars, variable-length fatty acids, and other moieties modifying them, which form different species of development-related ascaroside species.”
“see Figure 2. The figure presents a schematic diagram showing ascarylose sugars, variable-length fatty acids, and other moieties modifying them, which form different species of mating-relevant ascaroside species.”
“see Figure 3. The figure presents a schematic diagram showing ascarylose sugars, variable-length fatty acids, and other moieties modifying them, which form different species of aggregation-related ascaroside species.”
“see Figure 4. The figure presents a schematic diagram 445 showing ascarylose sugars, variable-length fatty acids, and other moieties modifying 446 them, which form different species of other function-related ascaroside species.”
“Figure 5. Pheromones produced by a wide range of nematode species. The black part of the diagram indicates the indicated phylogenetic tree, which was constructed through the comparison of 18S ribosomal RNA (18S) gene sequences obtained from GenBank. The green part indicates the lifestyle of the nematode. The blue part indicates the pheromones secreted by the corresponding nematode, where “{“ indicates that both rows of compounds it contains are pheromones secreted by C. elegans.”
Lastly, the Reference list contains a lot of formal/typing inaccuracies (also, it might be updated in relation to ciliate pheromones that have recently been revised, e.g. in Alimenti et al 2022, JEM, and in Luporini et al 2016, EJP)
